# Adverse Reactions to Anti-SARS-CoV-2 Vaccine: A Prospective Cohort Study Based on an Active Surveillance System

**DOI:** 10.3390/vaccines10030345

**Published:** 2022-02-23

**Authors:** Emanuele Amodio, Giuseppa Minutolo, Alessandra Casuccio, Claudio Costantino, Giorgio Graziano, Walter Mazzucco, Alessia Pieri, Francesco Vitale, Maurizio Zarcone, Vincenzo Restivo

**Affiliations:** 1Department of Health Promotion, Mother and Child Care, Internal Medicine and Medical Specialties, University of Palermo, 90127 Palermo, Italy; emanuele.amodio@unipa.it (E.A.); giuseppa.minutolo@unipa.it (G.M.); alessandra.casuccio@unipa.it (A.C.); claudio.costantino@unipa.it (C.C.); walter.mazzucco@unipa.it (W.M.); francesco.vitale@unipa.it (F.V.); 2Unità Operativa Complessa di Epidemiologia Clinica con Registro Tumori, Azienda Ospedaliera Universitaria Policlinico “Paolo Giaccone”, 90127 Palermo, Italy; giorgio.graziano@policlinico.pa.it (G.G.); alessia.pieri@policinico.pa.it (A.P.); maurizio.zarcone@policlinico.pa.it (M.Z.)

**Keywords:** adverse reaction, vaccine, coronavirus, SARS-CoV-2, m-RNA, female, young people, active surveillance system

## Abstract

To date, Coronavirus disease (COVID-19) has caused high morbidity and mortality worldwide. To counteract the pandemic scenario, several vaccines against the etiological factor of severe acute respiratory syndrome coronavirus 2 (SARS-CoV-2) were developed and tested. At the end of December 2020, BNT162b2 (Comirnaty, Pfizer-BioNTech) was the first and only authorized vaccine in Italy for selected categories, such as healthcare workers, fragile patients and people aged over 80 years old. To master our knowledge about BNT162b2 adverse reactions (ARs), an active surveillance system based on instant messaging was realized for voluntary participants who had been vaccinated at COVID-19 Vaccination Center of the Palermo University Hospital. Overall, 293 vaccinated persons were included in this study, which were more frequently healthcare workers (*n* = 207, 70.6% with a median age of 36 years, IQR = 29–55) followed by health professional students (*n* = 31, 10.6% with a median age of 27 years, IQR = 25–29), reporting 82.6% of at least one local or systemic AR. In details, the frequency of at least one local or systemic AR after the second dose of Comirnaty (*n* = 235, 80.2%) was statistically significant with higher value in comparison to the first one (*n* = 149, 50.9%; *p* < 0.001). However, local pain, swelling, joint pain and muscular pain after the second dose were the symptom causing a statistically significant working limitation. The youngest persons showed a higher risk to have either local or systemic ARs (aOR = 7.5, CI 95% = 2.9–18.9), while females had a higher risk of having systemic ARs (aOR = 1.8, CI 95% = 1.1–3.0). Despite the small sample examined, this active surveillance system by instant messaging seems to detect a higher ARs prevalence with respect to data obtained by the passive surveillance. Further studies could be required in order to optimize this clinical monitoring that could be considered an efficient and timely active surveillance.

## 1. Introduction

Coronavirus disease 2019 (COVID-19) is a life-threatening condition [1,2] caused by severe acute respiratory coronavirus 2 syndrome (SARS-CoV-2) [1], being worldwide lethal in ~2% of cases [2]. To reduce its morbidity and mortality, the most efficacious prevention strategy is vaccination [2]. Healthcare workers are one of the principal targets of vaccination because of their occupational exposure [2,3].

Nowadays, at least 300 vaccines against SARS-CoV-2 are being experimented [4]. However, the first available to be administered in Italy was BNT162b2 (Comirnaty, Pfizer-BioNTech) [2], whose main constituent is the mRNA encoding the SARS-CoV-2 spike protein. For its pharmacological characteristics, BNT162b2 demonstrated high immunogenicity and tolerability in clinical trial studies [5]. Although the immunization against SARS-CoV-2 with BNT162b2 was effective in 95% of the enrolled participants during the phase 3 of clinical trials [6,7], knowledge about adverse reactions (ARs) in real-world experiences should be improved [8]. Ideally, the pharmacovigilance system in place should improve vaccination compliance by contrasting the fear of ARs due to vaccines and forecasting the real weight of less severe ARs [9]. Among the several ways to individuate any AR suddenly [8], passive surveillance systems receive the ARs signalizations which either general population or healthcare workers can edit [10,11,12]. However, this method might underestimate the real frequency of ARs among vaccinated persons [11,13]. According to available literature, a passive surveillance system may have underestimated about one-third of all potential ARs after drugs assumption [12]. A recent study used a passive surveillance system based on a telephone application for registering ARs due to SARS-CoV-2 vaccination [6], whose data about ARs frequency compared to the clinical trials were quite different [6]. On the other hand, the main characteristic of an active surveillance system is to survey ARs directly during the follow-up period [9,13]. The ways to conduct an active surveillance can be various such as questionnaires by telephone [2,12,13], short message service (SMS) [9,14], and other informative systems [12]. An active surveillance had been created in Republic of Korea [14], where the frequency of either local (72.7%) or systemic ARs (86.5%), after the second dose of BNT162b2, were higher than those reported by other passive surveillance systems [6,14]. A cross-sectional study conducted by a telephone questionnaire to report the ARs arising in Italian healthcare workers after the completed vaccination schedule of BNT162b2 founded out at least one AR in 82% [2]. Another Italian research on healthcare workers emphasized the effectiveness of an active ARs surveillance performed by the hospital pharmacy showing women and individuals under 50 years old to have a higher predisposition to ARs [1]; nevertheless, the percentage of ARs after either the first or the second dose of BNT162b2 seemed to be changeable and undefined [1].

In Italy, the passive surveillance of ARs is provided by the National Network of Pharmacovigilance ruled by the national drug agency (AIFA) [2,10,13], while to date no national active ARs surveillance system is in place [2,13].

Since December 27, 2021, BNT162b2 was the first anti-SARS-CoV-2 vaccine available in Italy [2]. This vaccine included two doses of 30 μg with the interval time of at least 21 days [15]. According to the National Strategic Plan for anti-SARS-CoV-2 vaccination, all healthcare workers, including physicians, nurses, administrative personals, and all other workers into private or public healthcare settings, were one of the first targets to receive the vaccine against COVID-19, together with fragile hospitalized people and subjects aged over 80 years old [8].

The aim of this study is to evaluate the use of an active ARs surveillance based on instant text messages as a method for a timely identifying of ARs, including those with a mild presentation.

## 2. Materials and Methods

A prospective cohort study was conducted to recruit vaccinated persons accessing the COVID-19 Vaccination Center at the Palermo University Hospital “Paolo Giaccone” (UHP), from 21 January 2021 to 20 April 2021. All personal data were managed according to the Italian regulations [16,17]. Palermo 1 Ethic Committee approved the study on 20 January 2021.

### 2.1. Study Population

Following the guidelines established for the UHP [18,19], between 7 January 2021 and 31 March 2021, all the employees or training healthcare workers recipients were divided into the following priority vaccination categories:-employees in the period 28 December 2020 to 17 February 2021 [18,19];-medical residents attending specialty schools at the Medical School of Palermo University in the period 15 January to 13 February 2021 [18];-undergraduates students joined to the Medical School at University of Palermo in the period 10 February–31 March 2021 [19].

Since 20 February 2021, all people aged over 80 years old belonging to general population, which also includes people with comorbidities, could also receive vaccination due to higher risk of having a severe COVID-19 outcome [20,21].

All categories of people with vaccine availability in the UHP Vaccination Center were eligible for study inclusion.

### 2.2. Inclusion Criteria

All persons aged over 18 years old and vaccinated at the Vaccination Center were enrolled in this active surveillance-based study, between 21 January and 20 April 2021. After reading the informative consent which reported the instructions to participate and the way of personal data management [16,17,22], potential participants could adhere totally voluntary, when they received either the first or the second dose, by sending to the Vaccination Center phone number a confirmation of their participation with a short text message, which reported their own Italian identification code and the confirmation of their willingness to participate to the study. All the potential participants unable to send text messages, as well as all persons refusing to give the own personal data, persons without matching demographic variables (sex, age, and occupational status) and AR diaries for the first and second doses were excluded.

### 2.3. Active Surveillance System

The active surveillance system was based on a daily sending of a questionnaire through WhatsApp messenger (2021 © WhatsApp LLC, Menlo Park, CA, USA). WhatsApp is an Internet-based instant messaging system for any mobile devices, in which each message sending is protected from any kind of interception by third parties according to an end to end cryptography mechanism [23]. According to previous experiences conducted in different healthcare settings and to cheapness and personal data preservation [23,24,25], it was assumed that the use of WhatsApp instant messaging could improve communication between healthcare professionals and patients with ARs and reassure them about treatment of ARs and acceptability of vaccines.

### 2.4. Online Questionnaire

The questionnaire was created by using Google Form (©2021 Google, Mountain View, CA, USA) with the aim to report any AR of the ones listed and observed in the BNT162b2 trial [15]. The ARs questionnaire was divided in three parts: (1) presence of any signs/symptoms potentially associated to the vaccination; (2) local ARs such as pain, erythema, swelling, and others (that were all local ARs not mentioned before such as local pruritus or paresthesia); (3) systemic ARs, such as temperature over 37.5 °C, shivers, muscular pain, articular pain, headache, asthenia, lymph nodal swelling, nausea, and other systemic reactions (any general sign or symptom after vaccination which the vaccinated subjects could describe, such as diarrhea or dizziness).

Limitations to daily activities due to local or systemic ARs were measured by a Likert scale from 1 (ARs that were transitory and do not interfere daily activities) to 5 (ARs that interfere significantly with daily activities).

### 2.5. Follow-Up

The ARs diary consisted of seven questionnaires. Each of them was sent every day for a week after the first and the second dose, according to the time period of ARs surveillance in the BNT162b2 trial [15]. Only the persons recruited at the second dose received a summarizing questionnaire for ARs within 7 days after the first dose of BNT162b2. Participants were considered lost to follow-up whether they did not fill almost one daily questionnaire of ARs diary.

### 2.6. Statistical Analysis

Statistical analyses were performed with Stata/SE 14.2 (StataCorp LLC., College Station, TX, USA). The minimum sample size was calculated according to a predicted percentage of 80% ARs and a 5% precision in the estimated percentage. The requested sample size was 250 people, considering a 5% excess due to lost to follow up. During the study period, some participants were vaccinated with the first dose of BNT162b2 before 21 January 2021, therefore the denominator of this study was all vaccinated persons between the least recent data of the first dose and the most recent of the second one. BNT162b2 doses administered at the Vaccination Center from 28 December 2020 to 20 April 2021 were reported as total vaccinations consisting of the first and the second doses.

Absolute and relative frequencies were calculated for all demographic (sex, age, and occupational status) and vaccination data (vaccination date, range in days between the first and the second dose). After performing Skewness and Kurtosis test for quantitative variables, mean and standard deviation (SD) were chosen whether values were normally distributed, otherwise median and interquartile range (IQR) were selected. Following 33 and 66 percentile distribution, age was classified into three classes: under 31 years old, 31–45 years old, and over 45 years old. McNemar test for matched data evaluated the difference of signs and symptoms percentage after the second dose as compared to the first one. Wilcoxon test was performed to compare the median grade of daily activities limitation and duration in days of ARs at the second dose versus the first one. Bivariate analysis included all variables associated to have an ARs, such as sex, age class, occupational status as healthcare workers, and number of days between the first and the second BNT162b2 dose. Only the variables statistically significant in the bivariate analysis were included in the multivariate analysis. Crude and adjusted odds ratio (cOR and aOR, respectively) with related confidence interval at 95% (95%CI) were reported. All the results were considered statistically significant whether *p*-value was below 0.05.

## 3. Results

During the study period, 30,634 doses of BNT162b2 were administrated at the UHP Vaccination Center, of which 18,433 (60.2%) were first doses. Recipients of both two doses were 11,973 from 28 December 2020 to 20 April 2021.

Overall, 395 vaccinated individuals accepted to voluntarily participate to the study. After removing 102 respondents (25.8%) without matching diaries for the first and the second dose, 293 persons were included in the analysis (Figure 1). Among total enrolled population, 34.1% (100/293) were recruited since the first dose.

Table 1 shows demographic and vaccination characteristics of enrolled subjects. Most of them were males (*n* = 159, 54.3%) with a median age of 36 years old (IQR = 29–52 years old). Healthcare workers in the sample were 207 (70.6%), followed by healthcare students (10.6%, *n* = 31). Vaccinated persons receiving the second dose at the 21st day after the first one totaled 202 (68.9%).

Table 2 reported absolute and relative frequencies of ARs and the median of the maximum grade of daily activities limitation due to ARs. Overall, 242 (82.6%) participants reported at least 1 local or systemic ARs, of which the most frequent were pain (*n* = 227, 77.5%), as local ARs, and asthenia (*n* = 155, 52.9%), as systemic ARs. Vaccinated individuals who registered at least 1 local or systemic ARs after the second dose were 80.2% (*n* = 235), and 50.8% (*n* = 149) after the first dose (*p* < 0.001). Both local and systemic ARs after the second dose (respectively, 74.7% and 70.3%) were higher than after the first one (48.5% and 21.8%, respectively, *p* < 0.001). The most frequent reported local AR after the second dose was pain (74.1%, *n* = 217), which was almost twice than after the first dose (46.8%, *n* = 137, *p* < 0.001). The second most frequent local AR was swelling, which was also higher after the second dose (19.5%, *n* = 57) than the first one (12.6%, *n* = 37, *p* < 0.001). Asthenia was the prevalent systemic symptom after both doses of BNT162b2, even though it increased after the second dose (50.5%, *n* = 148) in comparison to the first one (13.3%, *n* = 39, *p* < 0.001). Other systemic ARs such as fever (>37.5 °C), joints and muscular pain, headache, and shivers increased after the second dose than the first one (*p* < 0.001). On the other hand, 2 (13.3%) persons referred allergic skin reactions after the first dose but not after the second one.

Grade of daily activities limitations due to ARs after the first and the second dose are reported in Table 2. Pain, swelling, joint pain, and muscular pain after the second dose (median value = 3) could interfere with daily activities more than the first one (median value = 2, *p* < 0.001, *p* = 0.0406, *p* = 0.0034, and *p* = 0.0154, respectively). The difference in daily activities limitations due to the other ARs registered at first and the second dose was not statistically significant.

Median duration in days of ARs was detected exclusively in vaccinated people recruited since the first dose (*n* = 100, 34.1%). The median duration of any local or systemic ARs was 2 days for the first dose (IQR = 1–2) and the second one (IQR = 1–3). The long lasting was local pain with a median of 2 days (IQR = 1–2), followed by swelling, shivers, and nausea with a median of 1 day (IQR = 1–2). The median duration difference of all ARs was not statistically significant between the two doses (data not shown in table).

The factor associated to have almost 1 local or systemic ARs in multivariate analyses (Table 3) was age lower than 31 years old (aOR = 7.5, CI 95% = 3.0–18.9). Furthermore, to have a local AR was associated to young age with an increasing trend (31–45 years old aOR = 2.0, CI 95% = 1.1–3.7 and <31 years old aOR = 4.6, CI 95% = 2.2–9.5). People who had at least a systemic AR were more probably female (aOR = 1.8, CI 95% = 1.1–3.0) and with an age lower than 31 years old (aOR = 3.3; CI 95% = 1.7–6.3).

## 4. Discussion

This study evaluated the feasibility of an active surveillance system by instant phone text messaging to early detect local and systemic ARs after BNT162b2 vaccination. The system proposed allowed it to document at least 1 local or systemic ARs in 82% of the respondents. Nearly all signs/symptoms increased statistically significant after the second dose in comparison to the first one, as already documented by previous studies [1,25]. More in depth, an Italian survey showed an increased risk of mild (213.0%) and serious (525.0%) ARs after the second dose of BNT162b2 [25].

Considering the total ARs after the completed vaccination schedule, the most frequent local AR was pain at the site of injection (77.5%), whose activity limitations were higher after the second dose. Moreover, the most frequent systemic AR was asthenia (52.9%) followed by muscular pain and headache (both 50.5%). The active surveillance conducted within the BNT162b2 clinical trial [6,7,15] detected as more frequent ARs local pain after the first and the second dose (77% vs. 72%), followed by headache (33.5% vs. 45.5%), and muscle pain (17.5% vs. 33%) [15]. All of these ARs following the first dose were slightly higher than those identified in this study, but they were much greater than findings from the passive surveillance [6]. In addition, the frequency of local and systemic ARs increased after the second dose in comparison to the first dose, similarly to other studies [14,15]. A previous Italian study showed that the second dose of BNT162b2 vaccine slightly increased the frequency of ARs in comparison to first dose (82.6% vs. 82.1%) [2]. The same results were obtained by Ossato et al., in which the second dose caused more ARs (69.7% vs. 31.7%, *p* < 0.001) [1]. The only exception was the study conducted by Menni et al., where local ARs following the first dose (71.9%) were more frequent than the second dose (68.5%, *p* < 0.0001) [6]. As reported by others, the main reason for the highest frequency of ARs after the second dose is the strengthening of the immune response against SARS-CoV-2, [1,26] leading to a generalized inflammatory status, involving humoral and cellular immunity [1] In detail, as well as empowering the antibody responses [1,26,27], the completed vaccination schedule increased also interferon γ (INF-γ) and other pro-inflammatory cytokine levels [1,27] which could be responsible for the onset of most systemic ARs due to a damage of local tissues. [1] During the immune activation after BNT162b2 vaccination, other cytokines as interleukin-2 (IL-2) were secreted [7,27]. In particular, some researches have already underlined febrile and asthenic effects of IL-2, as well as enhancing the production of other proinflammatory cytokines and mediators [28,29].

According to the National Pharmacovigilance Network, ARs after BNT162b2 schedule composed greater than 0.3% of 12,872,320 vaccinated subjects in Italy between 27 December 2020 and 26 April 2021 [10], whose relative frequency was lower than revealed in this study (0.3% vs. 65.5%) [10]. In addition, the percentage of identified ARs after the first and the second dose in this study was higher than the passive surveillance based on a mobile application proposed by Menni et al. for both local (78.2% vs. 66.2%) and the systemic ARs (73.0% vs. 25.4%) [6]. The main reason could be the passive reporting system of ARs used within the National Pharmacovigilance Network [10]. In this system, either vaccinated people or vaccinators could underestimate the importance to signalize even the mild ARs [11]. On the other hand, an active system could remind more than the passive the occurrence of any AR [11,12].

Several active ARs surveillance systems were reported in literature [14,25]. Whatever is the methodology used to detect any AR, an improved coverage of an ARs surveillance system or vaccinated persons’ adhesion to a ARs surveillance program might not correspond to an increase of signalizations. Although the active ARs surveillance based on an online survey by e-mail had detected a higher adhesion (24.5%) compared to this study (2.5%), the percentage of ARs after BNT162b2 schedule was lower (62.5% vs. 82.6%) [24]. Another study reported an active surveillance based on text messages such as short message service (SMS) or e-mail that had registered a quite lower frequency of local ARs after the second dose of BNT162b2 (72.7%) than that found in this study (74.7%), whereas the systemic ARs were higher (86.5% vs. 70.3%) [14]. Therefore, ARs frequency might change according to the biological characteristics of the population, comorbidity and other factors which regulate the immune systems [30,31]. On the other hand, the way of performing the active surveillance system could affect the ARs prevalence. Instant messaging as SMS increased the adhesion to the active ARs surveillance as compared to the telephone interviews, whose response rates were 90.1% and 66.7%, respectively [9]. A possible explanation might be the flexibility to the instant messaging reply, which could occur at any time, whereas the phone call answer cannot be postponed.

In this study, female sex was significantly associated to about a two-fold increased risk to have almost one systemic ARs after the second dose. This result was similar to an Italian online survey which observed a higher risk for both mild (65%) and serious (233%) systemic ARs among women [24]. This higher frequency of ARs has been observed also in a study conducted in the United Kingdom, in which the first dose of BNT162b2 caused more ARs in female (OR = 1.89, 95% CI 1.85–1.94) [6]. Furthermore, the National Network of Pharmacovigilance in Italy detected more ARs among females after any anti-SARS-CoV-2 vaccine [10]. Several hypotheses can explain this difference by sex. Women have a different hormonal system with a higher level of estrogens which empowered both B and T cell-mediated immunity, making it more efficiently against viruses [30,32]. Moreover, some vaccinations stimulate some genes of the X chromosome implicated into the B cell-mediated immune response [32]. Moreover, the ratio of lean and fat mass among females might affect the onset of ARs [33]. Consequently, the risk for any AR could be increased in vaccinated female due to a stronger and qualitative different vaccine immune response [30].

In this study, subjects aged lower than 31 years old had an increased risk for local and/or ARs and had reported more ARs after the second dose of BNT162b2. Several studies have already revealed more ARs among younger people [1,2,6,25]. The median age of vaccinated persons with at least one AR could change depending on the use of an active or passive surveillance, as well as the characteristics of vaccinated or enrolled population. As it could be deducted from the management of the categories accessing the UHP Vaccination Center [18,19], it included also students attending the courses of the Medical School, usually aged under 30 years old. Their participation to this study might decrease the median age of the overall recruited persons (36 years old). Whatever the age of vaccinated people, the literature showed increased immune reactions among younger individuals [1,26]. This was confirmed also by the National Pharmacovigilance Network during the period between 27 December 2020 and 26 April 2021 [10]. A reason could be the stronger immune response among younger subjects, showing generally higher antibody titles and cytokines levels [27]. On the other hand, the senescence of the immune system determined a reduction in immune responses after vaccination in the elderly [1,26]. Furthermore, several findings had found an impaired development of antibody response among elderly subjects due to a lower IL-2 levels. This condition [34] might decrease the risk for ARs [1,26].

Last, several limitations of the study should be considered. Some participants did not fill the questionnaire that was sent them on each observational day. Furthermore, this active surveillance based on an instant messaging system can have introduced a selection bias, excluding all potential participants without this type of phone availability, or including only those with a higher health literacy. Moreover, medical history was not considered among the factors correlated to ARs after anti-SARS-CoV-2 vaccination, and thus some confounding effect due to co-morbidities cannot be excluded [6,35]. Among the participants there could be some who had received corticosteroids in order to reduce the risk for uncontrolled immune responses after BNT162b2 vaccination. However, the previous conditions should have occurred in a very low percentage of vaccinated and they could have determined a negligible decrease in the frequency of ARs among pre-medicated persons [36].

Despite the several limitations, this active surveillance allowed us to have a real value of ARs after administration of vaccines and to have promptly detect ARs making it possible monitoring of recipients’ health status with a good cost–benefit balance. Furthermore, this analysis allows the assessment of factors associated to have an ARs after anti-SARS-CoV-2 vaccine administration, which can be useful in approaching the counselling of people in order to identify and adequately inform who are at higher risk of ARs.

## 5. Conclusions

The proposed active surveillance system by instant messaging system detected a higher percentage of ARs after vaccination schedule than the passive surveillance system in place in Italy. In our sample the youngest persons aged lower than 31 years old and the female had an increased risk to have ARs. According to the high compliance and feasibility of this type of surveillance, future studies should investigate the effectiveness and related costs of this active ARs surveillance system following vaccination as compared to the National Pharmacovigilance Network in order to provide more timely and accurate data.

## Figures and Tables

**Figure 1 vaccines-10-00345-f001:**
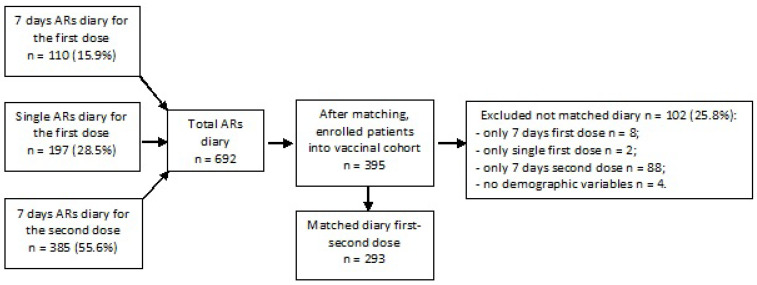
Flowchart of enrolled people in the study.

**Table 1 vaccines-10-00345-t001:** Demographic and vaccination characteristics of enrolled people.

Variables		*n* = 293
Sex (*n*, %)	Female	134 (45.7)
	Male	159 (54.3)
Age (median, IQR)		36 (29–52)
Age category in years (*n*, %)	>45	100 (34.1)
	31–45	93 (31.7)
	<31	100 (34.1)
Healthcare workers (*n*, %)		207 (70.6)
Healthcare students (*n*, %)		31 (10.6)
Frail people (*n*, %)		26 (8.9)
Administrative personnel (*n*, %)		26 (8.9)
Elderly (*n*, %)		3 (1.0)
Days between the first and the second dose (median, IQR)		21 (21–22)
Distance of days ≥ 21 (*n*, %)		91 (31.1)
Distance of days < 21 (*n*, %)		202 (68.9)

**Table 2 vaccines-10-00345-t002:** Absolute and relative frequencies of adverse reactions after the first and the second dose and completed vaccination schedule of BNT162b2.

Signs/Symptoms	First Dose (*n* = 293) N, %	Second Dose (*n* = 293) N, %	*p* *	Grade of Daily Activities Limitation after the First Dose Median, IQR	Grade of Daily Activities Limitation after the Second Dose Median, IQR	*p* **
**At least 1 localized/systemic**	149 (50.9)	235 (80.2)	<0.001	-	-	-
**At least 1 localized**	142 (48.5)	219 (74.7)	<0.001	-	-	-
Pain	137 (46.8)	217 (74.1)	<0.001	2 (1–3)	3 (2–4)	<0.001
Redness	9 (3.1)	19 (6.5)	0.0184	1 (1–3)	2 (1–3)	0.6698
Swelling	37 (12.6)	57 (19.5)	<0.001	2 (1–3)	3 (1–3)	0.0406
Other local reactions	17 (5.8)	19 (6.9)	0.7150	3 (3–3)	2 (2–4)	0.7815
**At least 1 systemic**	64 (21.8)	206 (70.3)	<0.001	-	-	-
Fever (>37.5 °C)	5 (1.7)	85 (29.0)	<0.001	4 (3–5)	4 (3–4)	0.3173
Shivers	21 (7.2)	131 (44.7)	<0.001	3 (2–4)	3 (2–4)	0.1172
Joint pain	25 (8.5)	137 (46.8)	<0.001	2 (2–4)	3 (2–4)	0.0034
Muscular pain	30 (10.2)	139 (47.4)	<0.001	2 (2–3)	3 (2–4)	0.0154
Headache	39 (13.3)	136 (46.4)	<0.001	3 (2–4)	3 (2–4)	0.3340
Asthenia	39 (13.3)	148 (50.5)	<0.001	3 (2–4)	3 (2–4)	0.0975
Lymphonodal swelling	7 (2.4)	31 (10.6)	<0.001	2 (1–3)	2 (2–3)	0.1025
Nausea	18 (6.1)	61 (20.8)	<0.001	2 (2–4)	3 (2–4)	0.0842
Other systemic reactions	18 (6.1)	46 (15.7)	<0.001	3 (2–4)	3 (2–4)	0.8689

* McNemar test, ** Wilcoxon test.

**Table 3 vaccines-10-00345-t003:** Bivariate and multivariate analysis for factors associated to ARs after vaccination against SARS-CoV-2 (first dose and second dose).

Dose		Any Sign or Symptom				Any Local Sign or Symptom				Any Systemic Sign or Symptom			
		cOR	95% CI	aOR	95% CI	cOR	95% CI	aOR	95% CI	cOR	95% CI	aOR	95% CI
First	Female (vs. Male)	1.57	0.99–2.49	-	-	1.55	0.98–2.47	-	-	1.99	1.11–3.57	-	-
	Age 31–45 (vs. > 45 years)	1.81	1.02–3.22	-	-	1.53	0.86–2.71	-	-	1.82	0.90–3.70	-	-
	Age < 31 (vs. > 45 years)	2.66	1.5–4.7	-	-	2.66	1.50–4.71	-	-	1.66	0.82–3.35	-	-
	Healthcare worker (vs. No Healthcare worker)	1.41	0.85–2.35	-	-	1.32	0.80–2.20	-	-	0.96	0.52–1.76	-	-
	Distance between the first and the second dose (for day increase)	1.16	0.70–1.90	-	-	1.14	0.70–1.88	-	-	0.99	0.54–1.80	-	-
Second	Female (vs. Male)	2.09	1.16–3.76	1.81	0.97–3.35	1.81	1.06–3.09	1.57	0.90–2.74	1.96	1.18–3.26	1.79	1.06–3.02
	Age 31–45 (vs. > 45 years)	2.15	1.11–4.15	1.92	0.98–3.78	2.15	1.15–4.01	1.98	1.05–3.73	1.68	0.93–3.05	1.51	0.82–2.76
	Age < 31 (vs. > 45 years)	8.07	3.20–20.32	7.47	2.95–18.91	4.89	2.37–10.08	4.59	2.21–9.52	3.54	1.84–6.81	3.27	1.68–6.34
	Healthcare worker (vs. No Healthcare worker)	1.25	0.67–2.31	-	-	1.05	0.59–1.87	-	-	1.06	0.61–1.84	-	-
	Distance between the first and the second dose (for day increase)	1.34	0.73–2.45	-	-	1.39	0.80–2.42	-	-	1.35	0.80–2.29	-	-

cOR = crude Odds ratio. aOR = Adjusted Odds Ratio.

## Data Availability

Data will be available after requesting the reason to the Correspondence author.

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
