# Peer review of "Adverse Reactions to Anti-SARS-CoV-2 Vaccine: A Prospective Cohort Study Based on an Active Surveillance System"

_vaccines, 2022, doi:10.3390/vaccines10030345_

Round 1

Reviewer 1 Report

This is a well conducted study with interesting results showing a higher rate of banal ARs in 2nd dose vaccinated subjects as already showed by several reports. However, the rate of ARs is much higher in this study, based on an instant messaging system (SMS). It also shows a higher AR risk in young and female vaccinated subjects. Tables, figures and references are OK. Discussion is also well conducted.

The interest is the proposed surveillance system and its high compliance rate and good feasibility. Considering that the size of the studied population is really too small, this should be re-written as a short paper or a correspondance to get the interest out of the instant messaging method.

Author Response

Reviewer 1

  1. This is a well conducted study with interesting results showing a higher rate of banal ARs in 2nddose vaccinated subjects as already showed by several reports. However, the rate of ARs is much higher in this study, based on an instant messaging system (SMS). It also shows a higher AR risk in young and female vaccinated subjects. Tables, figures and references are OK. Discussion is also well conducted.

The interest is the proposed surveillance system and its high compliance rate and good feasibility. Considering that the size of the studied population is really too small, this should be re-written as a short paper or a correspondance to get the interest out of the instant messaging method.

Thank you to Reviewer 1 for his/her suggestions and comments. We agree that the study includes a relatively small sample size however we would like to maintain its as full article since the manuscript has several reasons to be considered innovative and with a wide scientific appeal.

Firstly, the study is based on the feasibility of an active surveillance system with a short messages system. This assures a higher response to the signaling of adverse reactions, such as mild ARs, in comparison to the passive system where only severe adverse reactions are signalized. This can increase the rate of adverse reaction signalizations and this allow to not underestimate the real amount of adverse reactions as reported by Kim, T. et al in the Republic of Korea.

Secondly, it reveals the factors associated to adverse reactions after vaccination. In this manuscript factors associated to have an adverse reaction after anti-SARS-CoV2 vaccination are sex and age. This concept is useful in approaching the counselling of the people who want to have a SARS CoV2 vaccination in order to identify and adequately inform people who are at higher risk of adverse reactions.

These motivations suggest that shifting the manuscript from original article to short paper or correspondence could mind the elimination of some relevant data, as well as omitting information that could be useful for counselling of patients.

The section of discussion was modified to highlight the innovation of the manuscript as follows “Despite the several limitations, this active surveillance allowed us to have a real value of ARs after administration of vaccines and to have promptly detect ARs making it possible monitoring of recipients’ health status with a good costs-benefits balance. Furthermore, this analysis allows to assess factors associated to have an ARs after anti-SARS CoV2 vaccine administration that can be useful in approaching the counselling of the people in order to identify and adequately inform who are at higher risk of adverse reactions.”

Reviewer 2 Report

The paper is interesting and well written. However, I have to underlyne a point of interest:

  1. The authors do not describe if patients with autoimmune diseases are present in the cohort of vaccinated subjects
  2. I suggest to consider and add as reference the paper by Ferri et al concerning COVID19 and rheumatic diseases
  3. I suggest to consider and add as reference the position of Murdaca et al concernign vaccinations including vs Sars-Cov2 in systemic sclerosis

Author Response

The paper is interesting and well written. However, I have to underlyne a point of interest:

Q. The authors do not describe if patients with autoimmune diseases are present in the cohort of vaccinated subjects

A. Thank you for your interesting point of view. Unfortunately, we do not have information about comorbidities of vaccinated subjects. However, the patients with comorbidities as autoimmune diseases are more frequent among over 80 years old people for which vaccination was available after healthcare workers according to anti-SARS COV2 Italian vaccination plan. This clarification was also included in the methods section “Since February, 20 2021, all people aged over 80 years old belonging to general population, which include also people with comorbidities, could get vaccinated due to higher risk of having a severe COVID outcome. [20, 21]”.

Q.I suggest to consider and add as reference the paper by Ferri et al concerning COVID19 and rheumatic diseases

A. We included the suggested reference in the discussion methods section.

Q. I suggest to consider and add as reference the position of Murdaca et al concernign vaccinations including vs Sars-Cov2 in systemic sclerosis

A.I really appreciate your reference and added it in the discussion section to highlight the role of co-morbidities in vaccine ARs.

Round 2

Reviewer 1 Report

I still feel it is a good idea but not of long range

Author Response

The main reason to know the feasibility of this type of active surveillance is the ease of application in any vaccination center which want to detect ARs after any vaccination. This can aid to decrease vaccination hesitancy due to a better monitoring of health conditions after vaccination. Although the ARs active surveillance system based on instantaneous messages could be more time consuming than the passive one, it allow us to obtain a real amount of ARs data among population. This is extremely relevant for each healthcare professional who can give tailored personal counselling about the ARs before vaccination.

Furthermore, I should pay attention to a higher feasibility of an ARs active surveillance system at the national level, which can be applied for all kind of vaccines and other drugs, giving an immediate and cheap response about any adverse reaction. The knowledge of these ARs could improve better the management of vaccinated people who wanted to be reassured about the safety of vaccination. Moreover, other innovative studies published with a similar sample size such as Ripabelli et al. 2021 which included only 340 people used a cross sectional design. On the other hand, this study had a similar sample size but with a prospective design that was more solid in order to estimate the rate of adverse reactions. For all these reason, the study can be considered of a relevant scientific sound and a starting point to assess an active surveillance of vaccine through instantaneous messages.